

Emergent stationarity in Yellow River sediment transport and the
underlying shift of dominance: from streamflow to vegetation
Sheng Ye[1], Qihua Ran[1*], Xudong Fu[2], Chunhong Hu[3], Guangqian Wang[2], Gary
Parker[4], Xiuxiu Chen[1], Siwei Zhang[1]
[1] Institute of Hydrology and Water Resources, Department of Hydraulic Engineering,
Zhejiang University, Hangzhou 310058, China
[2] State Key Laboratory of Hydro-science and Engineering, Tsinghua University,
Beijing 100084, China
[3] State Key Laboratory of Simulation and Regulation of Water Cycle in River Basin,
Institute of Water Resources and Hydropower Research, Beijing 100048, China
[4] Department of Civil & Environmental Engineering and Department of Geology,
University of Illinois at Urbana-Champaign, Urbana, Illinois 61801, USA
[*] Corresponding author: Qihua Ran (ranqihua@zju.edu.cn)



**Abstract**

Soil erosion and sediment transport play important roles in terrestrial landscape
evolution and biogeochemical cycles of nutrients and contaminants. Although
discharge is considered to be a controlling factor in sediment transport, its correlation
with sediment concentration varies across the Yellow River Basin (YRB) and is not
fully understood. This paper provides analysis from gauges across the YRB covering
a range of climate, topographic characteristics and degree of human intervention. Our
results show that discharge control on sediment transport is dampened at gauges with
large mean annual discharge, where sediment concentration becomes more and more
stable. This emergent stationarity can be attributed to vegetation resistance. Our
analysis shows that sediment concentration follows a bell shape with vegetation index
(normalized difference vegetation index, NDVI) at annual scale despite heterogeneity
in climate and landscape. We obtain the counterintuitive result that as mean annual
discharge increases, the dominant control on sediment transport shifts from
streamflow erosion to vegetation retardation in the YRB.
**Keywords:** Yellow River Basin, sediment, stationarity, vegetation, bell-shape



## 1. Introduction

Watershed sediment transport, from hillslope to channel and subsequently the coast, is
crucial to erosion management, flood control, river delta development, and the
quantification of global biogeochemical cycles of materials such as organic
phosphorus, iron, and aluminum (Martin and Meybeck, 1979). During the 20th century,
human activities have significantly modified the landscape, leading to a reduction in
sediment yield and coastal retreat worldwide (Walling and Fang, 2003; Syvitski et al.,
2005). Known for its severe sediment problems, the Yellow River (YR) has been a
hotspot for studies on soil erosion and sediment transport for decades. Since the 1950s,
the annual sediment yield has reduced by 80% because of check dam construction and
ecosystem restoration such as the Grain-for-Green project, motivating discussion on
the necessity for further expansion of re-vegetation schemes (Chen et al., 2015).
Most studies on the physical mechanisms of soil erosion and sediment transport were
conducted in relatively small sub-catchments (Collins et al., 2004; Ran et al., 2012).
In order to interpret the patterns discovered at basin scale, then, it is essential to
understand the scaling effects of soil erosion and sediment transport. Specifically,
would the mechanisms identified at small scale also prevail at basin scale? If not,
what factors influence upscaling (Mutema et al., 2015; Song et al., 2016). However,
existing studies on the scaling effects of sediment transport are rather limited, and
show no significant spatial coherence in the scaling of sediment transport (Le
Bissonnais et al., 1998; Deasy et al., 2011; Song et al., 2016). Due to the great
heterogeneity in the YRB, scaling patterns could be different even within one tributary.



Taking the Wuding River as example, event mean concentration could decrease
downstream after the initial increase in one sub-catchment (Zheng et al., 2011) or
keep rising until reaching a plateau in another sub-catchment nearby (Fang et al.,
2008). Not only the sediment concentration, but also its correlation with discharge
varies across the YRB. Although discharge is considered as one of the controlling
factors in sediment transport, how its influence upscales remains to be fully
understood. Therefore it is necessary to expand our findings concerning sediment
transport from single tributaries to larger scales, especially incorporating diverse
climate, environmental and anthropogenic characteristics, so that we can derive an
understanding applicable to the whole YRB. In this paper, we collected observations
across the Yellow River Basin (YRB) to quantify changes in sediment concentration
in the recent decades (Rustomji et al., 2008; Miao et al., 2011; Wang et al., 2016). By
analyzing data from gauges across the YRB (Figure A1), we attempt to understand:
how the correlation between sediment concentration and discharge varies across
spatial and temporal scales; what are the dominant factors influencing sediment
transport in the YRB; and how their contributions vary from place to place.
**2.  Data and methodology**
We collected daily discharge and sediment concentration data from 123 hydrology
gauges within our study area: the YRB above Sanmenxia station, the major
hydropower station on the YR. From these we selected 68 gauges spanning a range of
climate conditions and physiographic areas, from the gauge at the most upstream end





of the main stem to the gauges above Tongguan, which just 100km upstream of
Sanmengxia Dam (Figure A1). These gauges were selected for at least 15-year (1971
– 1986) continuous daily discharge and sediment concentration records between 1951
and 1986. For comparison and further examination of our hypothesis, we also extract
the annual discharge and concentration data between 2000 and 2012 for seven gauges
located at the outlet of the major tributaries from the Yellow River Sediment Bulletin
(Figure A1 green stars).
The vegetation data used in this study corresponds to the normalized difference
vegetation index (NDVI) downloaded from NASA's Land Long Term Data Record
(LTDR) project, which provides daily NDVI observations globally at a spatial
resolution of 0.05˚. Instead of the NDVI obtained from Global Inventory Modeling
and Mapping Studies (GIMMS), LTDR is chosen for its better estimation in the YRB
(Sun et al., 2015). The daily NDVI data from 44 gauges located on the eight major
tributaries were collected and extracted according to the drainage area of the study
gauges from 1982 to 2012 (Figure A1 green stars). Annual maximum NDVI values
were used to represent the highest vegetation productivity. The precipitation and leaf
area index (LAI) data of the US catchments used for comparison are assembled from
the first author's previous work (Ye et al., 2015).
To examine the coupling between discharge and sediment concentration at various
temporal scales, wavelet coherence analysis was applied to the daily discharge ($m^3/s$)
and sediment concentration ($kg/m^3$) data following Grinsted et al (2004). Wavelet





transforms decompose time series into time and frequency and can be used to analyze
different parts of the time series by varying the window size. They have been applied
to geophysical records for the understanding of variability at temporal scales. To
examine the co-variation between discharge and concentration in the time frequency
domain, we used a wavelet coherence defined as (Grinsted et al 2004)
$$R^2(s) = \frac{\left| S(s^{-1} W^{XY}(s) \right|^2}{S(s^{-1} \left| W^X(s) \right|^2) \cdot S(s^{-1} \left| W^Y(s) \right|^2)} \qquad (1)$$

where $S$ is a smoothing operator, $W^{XY}$ is cross wavelet transform of time series X and
Y representing the common power between the two series, $s$ refers to scale and $W^X$
and $W^Y$ are the continuous wavelet transforms of time series X and Y respectively.
The wavelet coherence can be considered as a correlation coefficient of the two time
series in the time frequency domain. The region of cone of influence (COI) was
delineated in the wavelet coherence images to avoid reduction in confidence caused
by edge effects. Localized wavelets were also averaged through temporal scales to
obtain global wavelet coherence (Guan et al., 2011). More detailed explanation about
wavelet coherence analysis can be found in Grinsted et al (2004).
The discharge and the sediment yield (discharge x concentration) were aggregated
from daily to annually to further examine their correlation. This analysis is applied
only at annual scale since this is when the coupling from wavelet coherence analysis
is strongest. The annual mean concentration ($C_a$) was calculated by dividing the
annual sediment yield by annual discharge. The annual discharge ($Q_a$) and annual





mean concentration ($C_a$) was also averaged within the period 1951 to 1986 to obtain
the long-term mean annual discharge ($Q_m$) and the long-term mean annual
concentration ($C_m$). Note that both the parameters $Q_a$ and $Q_m$ used here are
area-specific discharges (mm/yr). For each gauge, a linear regression was fit to
describe the correlation between annual discharge ($Q_a$) and annual mean
concentration ($C_a$). The slope of this linear regression ($\alpha_{QC}$) is used to describe the
rate of change in sediment concentration with changing discharge at annual scale.
**3. The emergent stationarity in sediment concentration**
We applied wavelet coherence analysis to daily discharge and sediment concentration
data at 68 study gauges across the YRB (Figure A2, A3). The results show that the
coupling between discharge and concentration (Q-C) declines with mean annual
discharge ($Q_m$) at all three temporal scales (Figure 1a). That is, as $Q_m$ increases, the
influence of streamflow on sediment transport becomes weaker and weaker, both at
intra-annual and within-year scales.
This fading impact of streamflow as it increases can be further quantified in terms of a
linear regression between discharge ($Q_a$) and mean sediment concentration ($C_a$) at
annual scale, when the coupling between discharge and concentration (Q-C) is the
strongest (Figure A4). As can be seen from Figure1b, though annual mean
concentration is positively correlated with annual discharge at most gauges, the slope
in the Q-C regression ($\alpha_{QC}$) declines exponentially with $Q_m$ (*p*-value < 0.0001). The
larger $Q_m$ is, the less sensitive sediment concentration responds to variation in annual




discharge. For most gauges with $Q_m$ larger than 60mm/yr, $\alpha_{QC}$ is less than 0.1. When
$Q_m$ is larger than 100mm/yr, the variation in sediment concentration is less than 1% of
that in streamflow ($\alpha_{QC} < 0.01$), and thus sediment concentration can be approximated
as invariant to changing discharge.
This emergent stationarity explains the linear correlation between area-specific
sediment yield and runoff depth reported in a small sub-watershed in a hilly area of
the Loess Plateau (Zheng et al., 2013). Considering the sediment concentration to be
constant, the variation in yield is solely dominated by streamflow, resulting in the
observed linear discharge-yield relationship. Similar stationarity in sediment
concentration has also been found in arid watersheds in Arizona (Gao et al., 2013), US
where the sediment concentration becomes homogeneous among watersheds when
their drainage area is larger than 0.01 km$^2$. The difference in threshold for the
emergence of approximately discharge-invariant concentration between the YRB and
watersheds in Arizona, US is probably due to the differences in catchment
characteristics, i.e. vegetation type and coverage, terrestrial structure, soil properties,
etc.
Our analysis shows that mean annual discharge ($Q_m$) is a better indicator of the
correlation between water and sediment transport than drainage area, although the last
parameter has been used traditionally. Despite the heterogeneity, both the coupling
between Q-C and the concentration sensitivity to variation in streamflow decreases
with $Q_m$. A closer inspection reveals useful insights. At gauges with smaller values of





$Q_m$, discharge is the dominant factor in sediment transport: an increment in annual
discharge is amplified in the increment of sediment concentration ($\alpha_{QC} > 1$) (i.e.
Gauge 808, 812 in Figure A4). However, as $Q_m$ increases, variation in streamflow is
more weakly reflected in variation in sediment concentration, even though annual
mean concentration still correlates with annual discharge, (i.e. Gauge 806 in Figure
A4). As $Q_m$ continues to increase, sediment concentration becomes almost invariant to
discharge, suggesting that the dominant factor of sediment transport has shifted from
the discharge to something else.
**4.  The vegetation impact: a bell shape**
To further explore the potential cause of this emergent stationarity, we analyzed the
vegetation data (NDVI) from 44 of the gauges locating on eight major tributaries of
the YR (Figure A1). Our analysis shows that this declining sensitivity in concentration
at annual scale ($\alpha_{QC}$) is negatively related to vegetation impact (Figure 2).
For gauges with limited vegetation establishment in their drainage area, the variation
in discharge is amplified in sediment transport ($\alpha_{QC}>1$). The larger the discharge is at
specific year, the more sediment is eroded and mobilized per cubic meter. This
dominance of discharge is weakened when vegetation density and coverage increase.
Despite the larger sediment carrying capacity of larger discharge, sediment
concentration is reduced, probably due to the protection vegetation offers against
erosion. As maximum NDVI increase, sediment concentration becomes less and less
coupled with discharge at annual scale. When the vegetation density is sufficiently



high, sediment concentration is nearly stable in spite of the variation in discharge,
since the dense vegetation coverage protects soil from erosion and traps sediment.
That is, the emergent stationarity in sediment concentration corresponding to the
variation in discharge at gauges with large $Q_m$ can be attributed to the dampened
dominance of discharge due to the increasing impact of vegetation retardation.
To further confirm the vegetation impact on sediment transport, we derived the plot
between maximum NDVI and mean concentration at annual scale in Figure 3a. As we
can see, the annual mean sediment concentration follows a bell-shaped correlation
with vegetation establishment, with a peak concentration at a value of maximum
NDVI of around 0.36. On the falling limb of this bell curve, as NDVI increases, both
sediment concentration and $\alpha_{QC}$ decrease consistently. That is, both the value of
concentration and its sensitivity to streamflow variation declines with increasing
vegetation index on the falling limb. On the rising limb, however, both the value of
concentration and its sensitivity to streamflow variation increases with increasing
vegetation index. Most gauges have values $\alpha_{QC}$ larger than one, except one gauge
with an extremely small maximum value of NDVI. For these gauges, on the rising
limb, vegetal cover is still low in an absolute sense despite increasing NDVI.
Sediment concentration is mainly dominated by discharge: fluctuations in streamflow
are amplified in concentration ($\alpha_{QC}>1$). The only gauge with a value of $\alpha_{QC}$ smaller
than one is gauge HanJiaMao (HJM) at the Wuding River. Although the annual
precipitation and discharge at HJM is similar to other gauges along the Wuding River,
the annual mean sediment concentration is much smaller. This is because of the



extremely high baseflow contribution in discharge at HJM, which is around 90%,
thanks to very intensive check-dam construction there (Dong and Chang, 2014). Since
sediment in the YRB is mostly transported during large flow events during the
summer, smaller flow events are not capable of transporting significant sediment
loads at HJM.
In general, we can conclude that sediment transport is mainly dominated by discharge
when the vegetation index is low. With increasing NDVI, the impact of vegetation
grows slowly at first, and accelerates after the maximum NDVI exceeds 0.36.
Eventually, the effect of NDVI takes over the dominance of streamflow, and
attenuates the variation in sediment concentration (Figure 4). The nonlinear impact of
vegetation in regard to resistance of sediment to erosion is consistent with previous
findings (Rogers and Schumm, 1991; Collins et al., 2004; Temmerman et al., 2005;
Corenblit et al., 2009). When the vegetation index level is low, its resistance to soil
erosion develops slowly as vegetation grows and expands (Rogers and Schumm,
1991), and capability of vegetation to trap sediment is reduced when submerged by
flood (Temmerman et al., 2005) or overland flow. Therefore, for catchments with
limited vegetation establishment, the coverage of vegetation is insufficient to trap
sediment, nor is the vegetation able to protrude from the water level during the
extreme flow events that transport most of the sediment. Sediment transport in these
catchments is usually dominated by discharge. As NDVI increases, vegetation
becomes much more capable as an agent of erosion protection and sediment settling
(Jordanova and James 2003; Corenblit et al., 2009). With the compensation from



vegetation retardation, sediment and discharge become more and more decoupled as
discharge increases, so that concentration is nearly invariant to increasing discharge.
The transition point in maximum NDVI (around 0.36) is where the increment in
vegetation reduction balances with the incremental increase in water erosion. When
the capability of vegetation retardation catches up with streamflow erosion, the net
soil loss becomes negligible, a condition commonly observed in well-vegetated
regions.
**5. Validation of the bell shape across time and space**
Since 1999, a large-scale ecosystem restoration project, the 'Grain-for-Green' project
was launched in the YRB for soil conservation (Lv et al., 2012). It has substantially
improved vegetation coverage after a decade of implementation (Sun et al., 2015). To
validate our hypothesis gain from the early 1980s, we applied similar analysis to the
annual flow and sediment data as well as daily NDVI data at seven gauges located at
the outlets of major tributaries from 2008 to 2012 (Figure A1 green stars). This is the
period subsequent to the initiation of the 'Grain-for-Green' project. We have excluded
the years right after the implementation of the 'Grain-for-Green' project, when there
was an initial drastic change in vegetation coverage and sediment erosion and
transport processes.
As we can see from Figure 3b, there is significant increase in maximum NDVI for all
seven catchments, and considerable reduction in mean sediment concentration. This
improvement is consistent with the previous report that the 'Grain-for-Green' project





has made a remarkable achievement in regard to soil conservation in the YRB (Chen
et al., 2015). Comparison of the relationship between sediment concentration and
maximum NDVI in the early 1980s and around 2010 shows that the bell shape
relationship sustains even after drastic and significant anthropogenic alteration of the
land use and land cover across the whole YRB. Although the vegetation coverage has
improved significantly at all seven comparison gauges due to the ecosystem
restoration policy, and thereby effectively moderated sediment erosion; the bell shape
relationship between maximum NDVI and mean concentration sustains.
Similar bell shape relationship was also found for the multi-year mean annual
precipitation and sediment yield observed in the United States (Langbein and Schumm,
1958). The data used in the analysis of Langbein and Schumm (1958) was collected in
the 1950s from more humid and vegetated catchments with limited human
intervention, on the opposite of the YRB. Yet similar bell shape was still observed
between sediment yield and precipitation. Given the limited anthropogenic activities
in these catchments, vegetation growth is probably to correlate with annual
precipitation due to its adaption to climate, as in other US catchments (Figure A6).
Thus it is likely that a bell shape correlation between vegetation and sediment yield
would be found at these US catchments as well. This suggests that the bell shape
correlation between vegetation and sediment concentration is not only observed in the
YRB with intensive human intervention, but could also be valid outside it. More
analyses are needed to test this relationship in other catchments outside the YRB for
its universality.



## 6. Implications and conclusion

Our analysis shows that across the YRB, both the correlation between Q and C and
the magnitude of sediment response to the variation in streamflow decreases with $Q_m$.
When $Q_m$ is sufficiently large (i.e. > 60 mm/yr), sediment concentration reaches a
stationary (constant) state at annual scale. The emergent stationarity at gauges with
large $Q_m$ is related to the shift of dominance from discharge to vegetation. Because of
the slow development of vegetation resistance with increasing discharge for small
discharges, discharge dominates the soil erosion and sediment transport process until
the maximum NDVI exceeds a threshold (0.36 for this study), at which the parameter
governing concentration transits from streamflow erosion to vegetation retardation.
Our findings of the emergent stationarity in sediment concentration and the shift of
the dominant mechanism governing the Q-C relation have important implications for
water and sediment management at watershed scale. Our study indicates that for the
gauges with relatively large discharge, the annual mean concentration can be
approximated as a constant over a large range of discharges. Thus the estimation of
sediment yield can be simply inferred from a simulation of streamflow. First order
estimates of sediment yield for scientific or engineering purposes can be obtained by
multiplying the estimated discharge by a constant sediment concentration estimated
based upon the vegetation index. The correlation between vegetation and sediment
concentration will also be useful for the design of the ongoing ecosystem restoration
program known as the 'Grain-for-Green' project. The bell-shaped correlation between

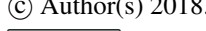



maximum NDVI and sediment concentration provides a quantitative way to estimate
the potential change in sediment concentration associated with proposed ecosystem
restoration planning schemes at and near each tributary. This can help guide land use
management so as to allocate the sediment contribution from each of the upstream
tributaries in a way that maintains the balance between erosion and deposition in the
lower YR.
It is important to collect more data from the current decade (i.e. after the substantial
ecosystem restoration) to further validate our findings in regard to emergent
stationarity and vegetation impact at more gauges in the YRB as well as other
watersheds worldwide. Numerical simulations are also needed to further explain the
detailed mechanism of vegetation retardation, including how it develops and how it
upscales.
**Acknowledgements**
This research was financially supported by the National Key Research and
Development Program of China (2016YFC0402404, 2016YFC0402406) and the
National Natural Science Foundation of China (51509218, 51379184, 51679209). All
the data used in this study were downloaded from websites indicated in Materials and
Methods section in Supplementary. The authors thank Dr. Jinren Ni for insightful
discussion.

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





**Figure 1:** Scatter plots between long-term mean annual discharge ($Q_m$) and (a)
wavelet $Q$-$C$ coherence at daily, monthly and annual scales, (b) slope of the discharge-
sediment concentration regression ($\alpha_{QC}$) at annual scale.

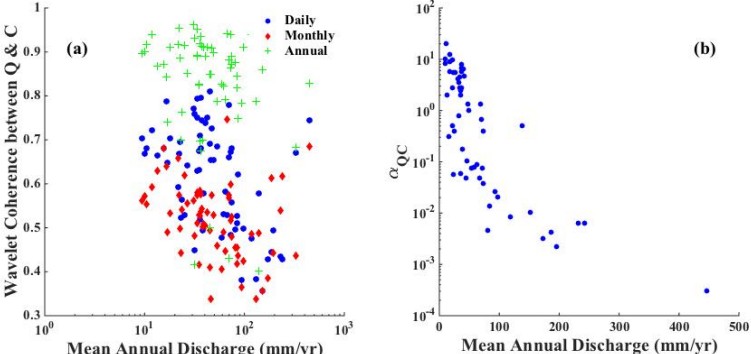







**Figure 2:** Scatter plots between the maximum NDVI and slope in the Q-C regression
($\alpha_{QC}$).

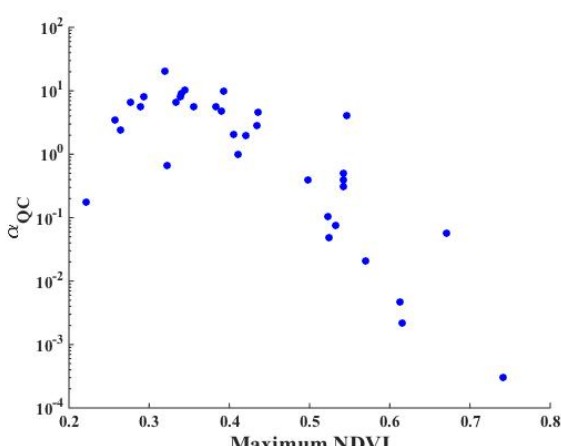






**Figure 3.** Scatter plot of annual mean concentration and maximum NDVI: (a) at 44
study gauges between 1982 and 1986, where the dots are color-coded by the slope in
the Q-C regression ($\alpha_{QC}$) at each gauge; and (b) at 7 gauges with both data from the
years 1982 – 1986 (blue dots) and the years 2008 – 2012 (red dots).

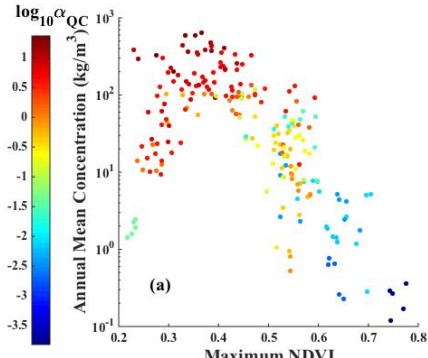
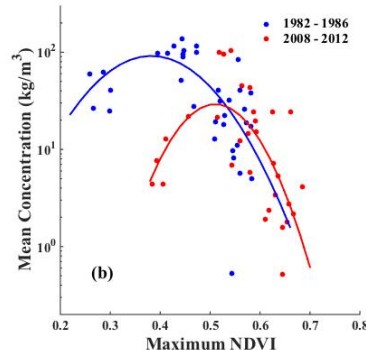





**Figure 4.** Illustration of the correlation between vegetation and sediment erosion,
retardation and the resulting sediment concentration in the YRB. Since vegetation
usually increases with discharge, with the rise in discharge, sediment eroded and
delivered by streamflow increases rapidly, while the retardation from vegetation is
limited at the beginning and increases fast afterwards. This non-synchronous impact on
sediment transport leads to the bell shape correlation between sediment concentration
and vegetation.

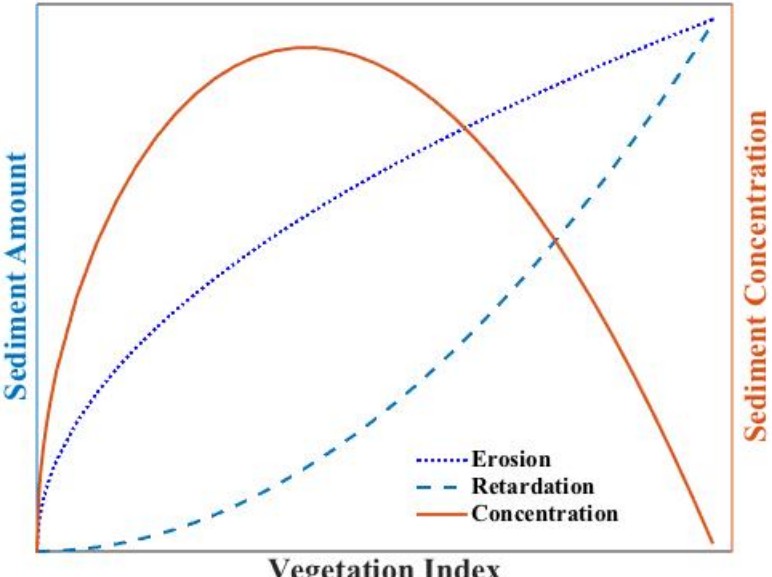








**Appendix**
**Figure A1:** Spatial distribution of hydrology gauges used in this study. The green
triangles correspond to 68 gauges with discharge and sediment concentration data, the
red triangles correspond to 44 selected gauges with NDVI data, and the green stars are
the ones with annual discharge and sediment data for the years 2000 – 2012.

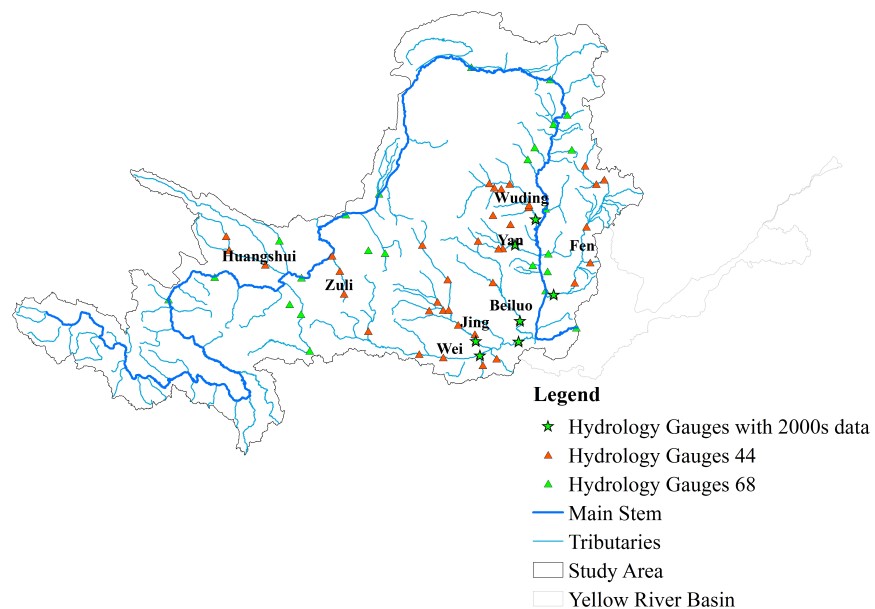






426 **Figure A2:** Wavelet coherence plots of the coupling between standardized discharge

427 and concentration, using the Jing River as an example. The labels correspond to the

428 gauge IDs. The shaded area is the cone of influence (COI) of edge effects.

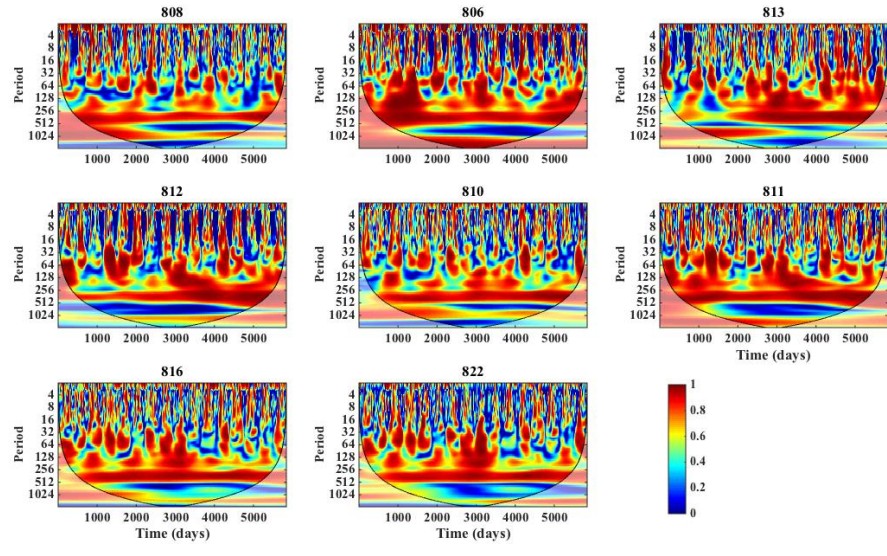




**Figure A3:** Averaged wavelet coherence plot, using the Jing River as an example. The
lines are colored according to long-term mean annual discharge (mm/yr), from blue to
brown as discharge increases.

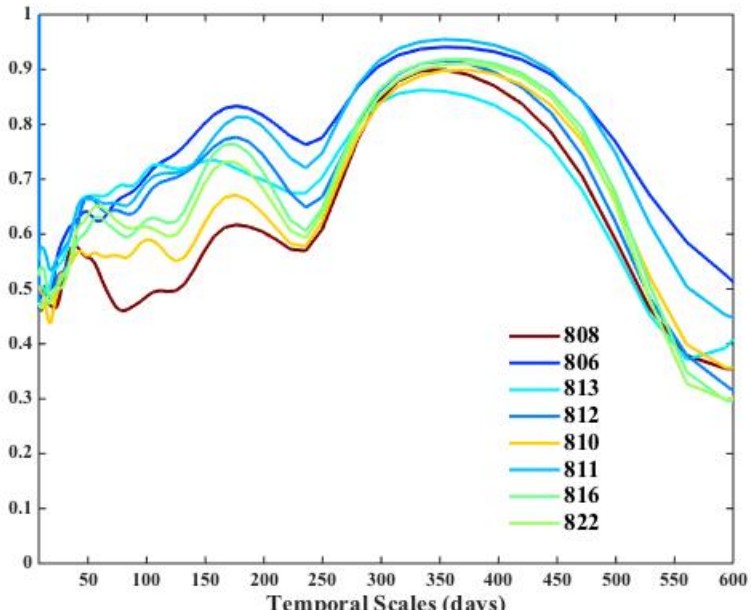





**Figure A4:** Scatter plot of the annual discharge and annual mean concentration from
1951 to 1986, as well as the result of linear regression between discharge and
concentration, using the gauges along the Jing River as an example.

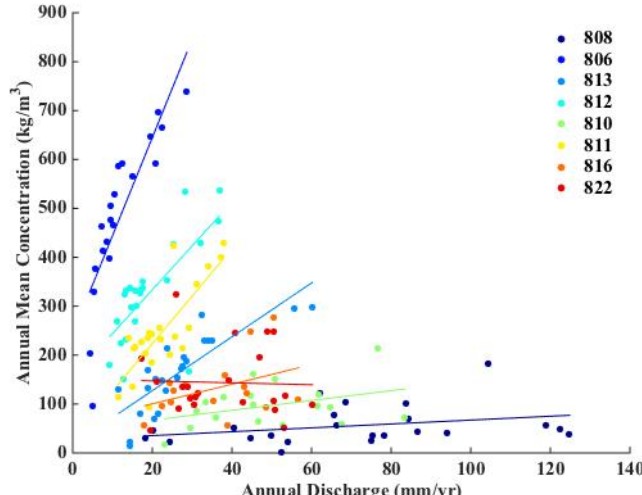




**Figure A5:** Spatial distribution of the slope of the Q-C regressions ($\alpha_{QC}$).

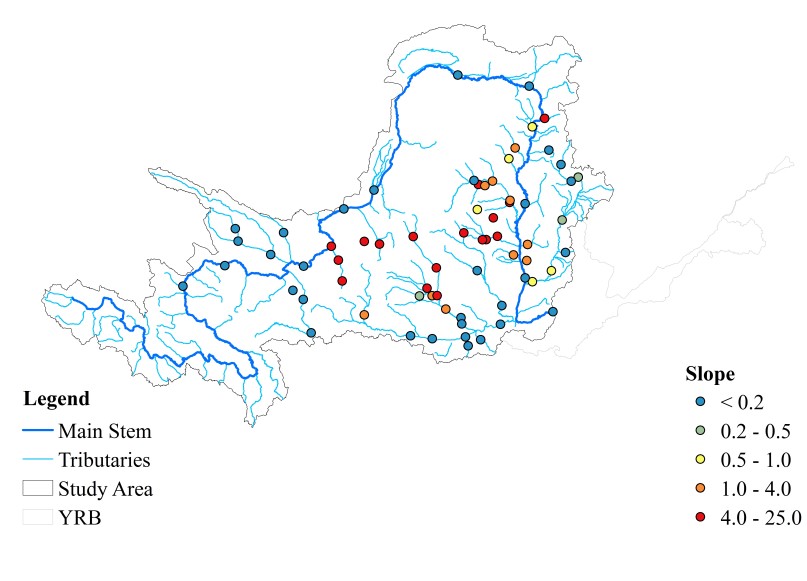






**Figure A6.** a) Spatial distribution of the MOPEX catchments; b) scatter plot of mean
annual precipitation and annual maximum LAI for the MOPEX catchments.

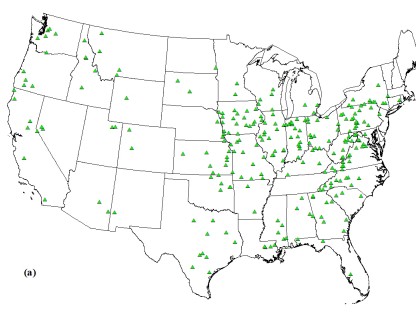

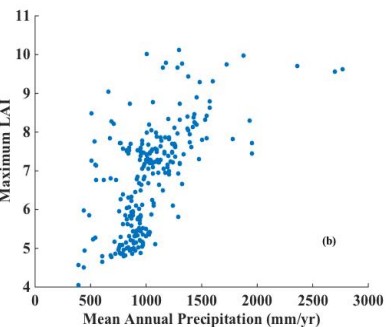
