# Peer review of "Emergent stationarity in Yellow River sediment transport and the"

_Hydrology and Earth System Sciences, 2018_

## Referee Comment (RC1) · Anonymous Referee #1 · 16 Aug 2018

The authors collected and analyzed hydrologic data to develop the relationships between sediment concentration and discharge, vegetation index and discharge and sediment concentration in the Yellow River Basin using Wavelet Coherence method. Eventually they drew some conclusions on these relationships. Both data and analysis well support these conclusions. The reviewer recommends to accept the paper with some minor revisions as follows, 1. Double check the whole manuscript and correct some typos such as: Line 108 (")" is expected in Eqn 1), Line 121, "the" strongest. . . etc. 2. Lines 118 – 129, use formula instead of description to explain the physical meaning of these parameters. 3. Even though NVDI has been described in the cited literature, it will be more convenient for readers understand the effect of vegetation if

the authors can briefly explain the definition. 4. More discussion on the determination of threshold value of discharge is expected. 5. How will vegetation type, climate, and other watershed characteristics affect the conclusion? A short discussion will be helpful.

---

## Referee Comment (RC2) · Anonymous Referee #2 · 29 Oct 2018

General comments: The authors quantified the annual impacts of discharge and vegetation density on the sediment concentration at dozens of gauges over the Yellow River basin. The conclusion is that the dominant controlling factor of sediment shifts from discharge to the vegetation resistance with discharge increasing, which is interesting. Besides, the manuscript was well written. However, some problems about the details of the assumptions and method used (i.e. wavelet coherence analysis and regression fitness) are expected to be explained more clearly, as these details are very critical to the reliability as well as reasonableness of results associated with main conclusion.

Several detailed comments are listed as follows: (1) "The sediment concentration fol-

[Figure]

lows a bell shape with NDVI at annual scale" was summarized throughout the text (e.g., Line 12/193/253), while the log-transformation was used to sediment concentration data in Figures 2 and 3. As we know, the log transformation is non-linear, thus the bell shape in Figures 2 and 3 may depend on this transformation approach. (2) In Figure 3, the authors should give the mathematical expression of the fitted curve with bell shape. Is it of a polynomial form or something else? Moreover, the goodness of the fit is expected to be presented. (3) It's doubtable that the so-called emergent stationarity is attributed only to vegetation resistance. The physical connection between vegetation condition and sediment concentration is not as explicit as that between discharge and sediment concentration. In addition, the discharge and the vegetation were separately incorporated to consider the impact to sediment condition. So, the conclusion in this study is under very strong assumption, i.e., the sediment condition in a basin is only controlled by discharge and vegetation. However, this assumption was not listed clearly. On the other hand, was it reasonable? In addition to vegetation, resistance of sediment to erosion may be related to other property of the basin, such as soil properties. Authors said sediment concentration follows a bell shape with vegetation index. I guess that the mean sediment concentration also follows a bell shape with the mean runoff. According to literatures, the discharge of 1982-1986 in many sub-basins of Yellow River was much larger than that in 2008-2012, how to compare the decreased discharge contribution with the increased NDVI contribution to the concentration? Can the authors add the plot of mean sediment concentration against annual discharge with the same period and gauges in both Figure 3a and 3b? (4) On line 92-96 about the NDVI data used, how area-specific NDVI was obtained from the spatial imagery? Since the downloaded imagery was global, why only NDVI at 44 gauges not the maximum 68 gauges was estimated. (5) Why not use the mean NDVI, but the maximum (daily?) NDVI, when you investigated the relationship between the NDVI and mean concentration at annual scale? How much uncertainty for the maximum NDVI exists? (6) In Figure A1, the text in legend is inappropriate, because there were not 68 green triangles plotted (maybe 68-44=22 gauges). (7) On line 120, "annual scale . . . is when

the coupling from wavelet coherence analysis is strongest", why? (8) On line 141-142, "slope in the Q-C regression (aQC) declines exponentially with Qm (p-value < 0.0001)". From the Figure 1b, it looks like log-transformation used to aQC. If so, then the text expression and plot were inconsistent. And the authors should give the mathematical equation of exponential decline trend and its fitted curve.

———————————————

---

## Author Comment (AC1) · 13 Nov 2018

Please check the supplement for the complete version of our response to reviewer 1
Anonymous Referee #1

The authors collected and analyzed hydrologic data to develop the relationships between sediment concentration and discharge, vegetation index and discharge and sediment concentration in the Yellow River Basin using Wavelet Coherence method. Eventually they drew some conclusions on these relationships. Both data and analysis

well support these conclusions. The reviewer recommends to accept the paper with some minor revisions as follows,

We appreciate the reviewer's insightful inputs that have helped to improve the quality of this manuscript. In response to the comments, we have made corresponding revisions. Our response to each comment is listed below in blue with the changes in manuscript, we also include the specific line numbers of the changes we have made. We hope the reviewer find the revision and responses sufficient.

1. Double check the whole manuscript and correct some typos such as: Line 108 (")" is expected in Eqn 1), Line 121, "the" strongest. . . etc.

We have corrected the typos as following, thank you (please see lines 116 and 133). L116: Rˆ2 (s)= ãĂŰ|S(sˆ(-1) WˆXY (s))|ãĂŮˆ2/(S(sˆ(-1) |WˆX (s)|ˆ2 )*S(sˆ(-1) ãĂŰ|WˆY (s)|ãĂŮˆ2)) L133: "This analysis is applied only at annual scale since this is when the coupling from wavelet coherence analysis is the strongest."

2. Lines 118 – 129, use formula instead of description to explain the physical meaning of these parameters.

We have now replaced the description of the parameters by equations as following, thank you (please see lines 126 – 144).

The annual discharge (Qa) and the sediment yield (La) were aggregated from daily to further examine their correlation: $Q\_a=(\sum\_(i = 1\Theta n(Q\_i * 3600 * 24))/Ad * 1000 (2) L\_a = (\sum\_(i = 1\Theta n(Q\_i * C\_i * 3600 * 24)) (3) where Qi(m3/s) and Ci(kg/m3) are the dai$

3. Even though NVDI has been described in the cited literature, it will be more convenient for readers understand the effect of vegetation if the authors can briefly explain the definition.

We have added following brief explanation on NDVI in the manuscript, we hope the reviewer find this satisfactory (please see lines 90 – 95).

The vegetation data used in this study corresponds to the normalized difference vegetation index (NDVI), which is an index calculated from remote sensing measurements to indicate the density of plant growth (Running et al., 2004). The NDVI data was downloaded from NASA's Land Long Term Data Record (LTDR) project, which provides daily NDVI observations globally at a spatial resolution of 0.05âŮę.

4. More discussion on the determination of threshold value of discharge is expected.

We obtained the threshold value of discharge by the slope in the Q-C regression (ïĄąQC), 60mm/yr is where most ïĄąQC is less than 0.1 while 100mm/yr is where most ïĄąQC is less than 0.01. Those gauges with larger mean annual discharge are the ones downstream of the major tributaries or along the main stem of YR. For these gauges, due to the larger drainage area, there is significant heterogeneity in the catchments. The region generates more discharge doesn't necessary contribute most in sediment yield (Figure S4), factors other than discharge may play important roles. This threshold discharge was also found in arid watersheds in Arizona though with quite different numbers. This divergence could be attributed to the different catchment characteristics like soil type, topography and so on. It would be interesting to further study the cause of the threshold discharge at these specific values, but this is above the scope of this work and we will pursue this in our follow-up studies. We have now added the following discussion in the manuscript. Hopefully the reviewer finds it sufficient (please see lines 166 – 172 and 346 - 348).

L166: For example, gauges with ïĄąQC less than 0.1 are the ones with Qm larger than 60mm/yr. When Qm is larger than 100mm/yr, the variation in sediment concentration is less than 1% of that in streamflow (ïĄąQC < 0.01), and thus sediment concentration can be approximated as invariant to changing discharge. Most of these gauges locate on the main stem or near the outlets of tributaries. This increased independence between sediment concentration and discharge may be attributed to the heterogeneity in these relatively large catchments.

L346: Analysis with more field measurements could also help explain the threshold discharge of the emergent stationarity.

5. How will vegetation type, climate, and other watershed characteristics affect the conclusion? A short discussion will be helpful.

The vegetation types in the YRB include bare soil, grassland, shrubs and forest (Zhang et al., 2016), our conclusion is derived from these various vegetation types. But we only look at the NDVI in this study, it is possible that the capability to prevent soil erosion may vary with vegetation species despite of similar NDVI values. This worth exploring with more detailed studies in the future. On the other hand, the climate in the YRB is semi-arid and arid (mean annual precipitation varies within the range of 100mm to 800mm), it would be interesting to see whether our conclusion would sustain under humid climate. Although catchments with humid climate usually have well-developed vegetation coverage, thus the soil erosion issue is less severe, there could still be soil erosion problems. Thus, it would be interesting to study the soil erosion issue in those humid catchments. We have included the following discussion on this in the manuscript, we hope the reviewer will be satisfied with it (please see lines 342 – 346).

It will be helpful if we could examine our findings in other watersheds worldwide with different climate and vegetation types. Although humid regions are usually considered as well-vegetated, study shows that there could still be erosion issues in these areas due to topographic gradient, precipitation intensity, and soil properties, etc. (Holz et al., 2015).

Please also note the supplement to this comment:
https://www.hydrol-earth-syst-sci-discuss.net/hess-2018-265/hess-2018-265-AC1-supplement.pdf

**Supplement:**

The authors collected and analyzed hydrologic data to develop the relationships between sediment concentration and discharge, vegetation index and discharge and sediment concentration in the Yellow River Basin using Wavelet Coherence method. Eventually they drew some conclusions on these relationships. Both data and analysis well support these conclusions. The reviewer recommends to accept the paper with some minor revisions as follows,

We appreciate the reviewer's insightful inputs that have helped to improve the quality of this manuscript. In response to the comments, we have made corresponding revisions. Our response to each comment is listed below in blue with the changes in manuscript, we also include the specific line numbers of the changes we have made. We hope the reviewer find the revision and responses sufficient.

1. Double check the whole manuscript and correct some typos such as: Line 108 (")" is expected in Eqn 1), Line 121, "the" strongest. . . etc.

We have corrected the typos as following, thank you (please see lines 116 and 133).

L116: $R^2(s) = \dfrac{\left|S\left(s^{-1}W^{XY}(s)\right)\right|^2}{S\left(s^{-1}|W^X(s)|^2\right)*S(s^{-1}|W^Y(s)|^2)}$

L133: "This analysis is applied only at annual scale since this is when the coupling from wavelet coherence analysis is the strongest."

2. Lines 118 – 129, use formula instead of description to explain the physical meaning of these parameters.

We have now replaced the description of the parameters by equations as following, thank you (please see lines 126 – 144).

The annual discharge ($Q_a$) and the sediment yield ($L_a$) were aggregated from daily to further examine their correlation:

$$Q_a = \left(\sum_{i=1}^{n}(Q_i * 3600 * 24)\right)/Ad * 1000 \qquad (2)$$
$$L_a = \left(\sum_{i=1}^{n}(Q_i * C_i * 3600 * 24)\right) \qquad (3)$$

where $Q_i$ (m³/s) and $C_i$ (kg/m³) are the daily discharge and sediment concentration, $Ad$ is the drainage area (km²) of each gauge, $n$ is the number of days in each year. This analysis is applied only at annual scale since this is when the coupling from wavelet coherence analysis is the strongest. The annual mean concentration ($C_a$) was calculated as:

$$C_a = L_a/(Q_a * Ad/1000) \qquad (4)$$

The long-term mean annual discharge ($Q_m$) and the long-term mean annual concentration ($C_m$) was also calculated by averaging for the period of 1951 to 1986. Note that both the parameters $Q_a$ and $Q_m$ used here are area-specific discharges (mm/yr). For each gauge, a linear regression was fit to describe the correlation between annual discharge ($Q_a$) and annual mean concentration ($C_a$). The

slope of this linear regression ($\alpha_{QC}$) is used to describe the rate of change in sediment concentration with changing discharge at annual scale.

3. Even though NVDI has been described in the cited literature, it will be more convenient for readers understand the effect of vegetation if the authors can briefly explain the definition.

We have added following brief explanation on NDVI in the manuscript, we hope the reviewer find this satisfactory (please see lines 90 – 95).

The vegetation data used in this study corresponds to the normalized difference vegetation index (NDVI), which is an index calculated from remote sensing measurements to indicate the density of plant growth (Running et al., 2004). The NDVI data was downloaded from NASA's Land Long Term Data Record (LTDR) project, which provides daily NDVI observations globally at a spatial resolution of $0.05°$.

4. More discussion on the determination of threshold value of discharge is expected.

We obtained the threshold value of discharge by the slope in the Q-C regression ($\alpha_{QC}$), 60mm/yr is where most $\alpha_{QC}$ is less than 0.1 while 100mm/yr is where most $\alpha_{QC}$ is less than 0.01. Those gauges with larger mean annual discharge are the ones downstream of the major tributaries or along the main stem of YR. For these gauges, due to the larger drainage area, there is significant heterogeneity in the catchments. The region generates more discharge doesn't necessary contribute most in sediment yield (Figure S4), factors other than discharge may play important roles. This threshold discharge was also found in arid watersheds in Arizona though with quite different numbers. This divergence could be attributed to the different catchment characteristics like soil type, topography and so on. It would be interesting to further study the cause of the threshold discharge at these specific values, but this is above the scope of this work and we will pursue this in our follow-up studies. We have now added the following discussion in the manuscript. Hopefully the reviewer finds it sufficient (please see lines 166 – 172 and 346 - 348).

L166: For example, gauges with $\alpha_{QC}$ less than 0.1 are the ones with $Q_m$ larger than 60mm/yr. When $Q_m$ is larger than 100mm/yr, the variation in sediment concentration is less than 1% of that in streamflow ($\alpha_{QC}$ < 0.01), and thus sediment concentration can be approximated as invariant to changing discharge. Most of these gauges locate on the main stem or near the outlets of tributaries. This increased independence between sediment concentration and discharge may be attributed to the heterogeneity in these relatively large catchments.

L346: Analysis with more field measurements could also help explain the threshold discharge of the emergent stationarity.

5. How will vegetation type, climate, and other watershed characteristics affect the conclusion? A short discussion will be helpful.

The vegetation types in the YRB include bare soil, grassland, shrubs and forest (Zhang et al., 2016), our conclusion is derived from these various vegetation types. But we only look at the NDVI in this study, it is possible that the capability to prevent soil erosion may vary with

vegetation species despite of similar NDVI values. This worth exploring with more detailed studies in the future. On the other hand, the climate in the YRB is semi-arid and arid (mean annual precipitation varies within the range of 100mm to 800mm), it would be interesting to see whether our conclusion would sustain under humid climate. Although catchments with humid climate usually have well-developed vegetation coverage, thus the soil erosion issue is less severe, there could still be soil erosion problems. Thus, it would be interesting to study the soil erosion issue in those humid catchments. We have included the following discussion on this in the manuscript, we hope the reviewer will be satisfied with it (please see lines 342 – 346).

It will be helpful if we could examine our findings in other watersheds worldwide with different climate and vegetation types. Although humid regions are usually considered as well-vegetated, study shows that there could still be erosion issues in these areas due to topographic gradient, precipitation intensity, and soil properties, etc. (Holz et al., 2015).

---

## Author Comment (AC2) · 13 Nov 2018

Please check the supplementary file for our complete response to reviewer 2. Anonymous Referee #2

General comments: The authors quantified the annual impacts of discharge and vegetation density on the sediment concentration at dozens of gauges over the Yellow River basin. The conclusion is that the dominant controlling factor of sediment shifts from discharge to the vegetation resistance with discharge increasing, which is interesting.

Besides, the manuscript was well written. However, some problems about the details of the assumptions and method used (i.e. wavelet coherence analysis and regression fitness) are expected to be explained more clearly, as these details are very critical to the reliability as well as reasonableness of results associated with main conclusion.

We appreciate the reviewer's comments and have made our efforts to explain our assumptions and method used in the manuscript as the reviewer suggested. Our response to each comment is listed below in blue with the changes in manuscript, we also include the specific line numbers of the changes we have made. We hope the reviewer find the revision and responses satisfactory.

Several detailed comments are listed as follows:

(1) "The sediment concentration follows a bell shape with NDVI at annual scale" was summarized throughout the text (e.g., Line 12/193/253), while the log-transformation was used to sediment concentration data in Figures 2 and 3. As we know, the log transformation is non-linear, thus the bell shape in Figures 2 and 3 may depend on this transformation approach.

We agree with the reviewer that log-transformation would change the shape of the correlation between NDVI and concentration. But as we shown here the increase and decrease trend of the bell shape sustains and is clear in linear scale. As the concentration covers a large range from 0.1kg/m3 to 700kg/m3, the points with small concentration (i.e. <=100kg/m3) would all collapse. The differences among these points cannot be shown clearly in linear scale. Thus, to make the relationship clearer we choose the log-transformation for better presentation. We hope the reviewer finds our explanation satisfactory.

Figure 2R1: Scatter plots between the maximum NDVI and slope in the Q-C regression (ïĄąQC).

Figure 3R1: Scatter plot of annual mean concentration and maximum NDVI: (a) at 44
study gauges between 1982 and 1986, where the dots are color-coded by the slope in the Q-C regression (ïĄąQC) at each gauge; and (b) at 7 gauges with both data from the years 1982 – 1986 (blue dots) and the years 2008 – 2012 (red dots).

(2) In Figure 3, the authors should give the mathematical expression of the fitted curve with bell shape. Is it of a polynomial form or something else? Moreover, the goodness of the fit is expected to be presented.

Yes, it is a polynomial form. We have now shown the mathematical expression of the fitted curve, as well as the goodness of the fit in the figure. Thank you for your suggestion (please see the updated Figure 3).

Figure 3R2: Scatter plot of annual mean concentration and maximum NDVI: (a) at 44 study gauges between 1982 and 1986, where the dots are color-coded by the slope in the Q-C regression (ïĄąQC) at each gauge; and (b) at 7 gauges with both data from the years 1982 – 1986 (blue dots) and the years 2008 – 2012 (red dots). The R2 for the two fit is 0.6 and 0.44 respectively with p-value < 0.001 for both of them.

(3) It's doubtable that the so-called emergent stationarity is attributed only to vegetation resistance. The physical connection between vegetation condition and sediment concentration is not as explicit as that between discharge and sediment concentration. In addition, the discharge and the vegetation were separately incorporated to consider the impact to sediment condition. So, the conclusion in this study is under very strong assumption, i.e., the sediment condition in a basin is only controlled by discharge and vegetation. However, this assumption was not listed clearly. On the other hand, was it reasonable? In addition to vegetation, resistance of sediment to erosion may be related to other property of the basin, such as soil properties. Authors said sediment concentration follows a bell shape with vegetation index. I guess that the mean sediment concentration also follows a bell shape with the mean runoff. According to literatures, the discharge of 1982-1986 in many sub-basins of Yellow River was much larger than that in 2008-2012, how to compare the decreased discharge contribution with the increased NDVI contribution to the concentration? Can the authors add the plot of mean sediment concentration against annual discharge with the same period and gauges in both Figure 3a and 3b?

We are sorry about the confusion we made that "the sediment condition in a basin is only controlled by discharge and vegetation." What we are trying to say in this manuscript is that based on our findings of the correlation between NDVI and concentration from the data, vegetation plays an important role in soil erosion and sediment transport for all the study catchments. Combining with Figure 1 that the coupling between Q-C weakens with the increase in mean annual discharge, we have the results that: when mean annual discharge is small, both discharge and vegetation have good correlation with sediment concentration, while when mean annual discharge is relatively large, the correlation between vegetation and concentration sustains, but the correlation between Q-C fades out. For the former situation (mean annual discharge is small), Q-C is positively correlated, which is consistent with our intuitive that larger discharge delivered more sediment. While the positive correlation between NDVI and concentration is counterintuitive, as we usually think vegetation helps prevent soil erosion. Thus, we think for these catchments, the increase in concentration is caused by hydraulic erosion and transport. Although larger Q also enables growth in vegetation, the amount of vegetation coverage is not sufficient to resist soil erosion caused by discharge. That is, the correlation between NDVI and concentration for these gauges is not a causal relationship, but is more likely because of the discharge. On the other hand, for the latter condition when mean annual discharge is relatively large, the impact of discharge disappears while the resistance from vegetation takes the dominance. But as the reviewer pointed out that our understanding in physical connection between vegetation condition and sediment concentration is not as explicit as that between Q-C. Further studies on the physical impact of vegetation is essential to explain this bell shape correlation in the perspective of mechanism. Indeed, motivated by this finding, we have done numerical simulations on the change in soil properties like saturated hydraulic conductivity caused by re-vegetation in another manuscript forthcoming.

We have also studied the correlation between dominant soil types and sediment concentration, the plot is quite scatter, thus we didn't show it in the manuscript for brevity. The results is shown in Figure R1. It is possible that there are other factors we did't consider here that influences the sediment transport, however, given the good fit between maximum NDVI and concentration, it is reasonable to say that vegetation plays an significant role in the soil erosion and sediment transport in the YRB, though it may not be the only controlling factor.

Figure R1: Scatter plot of annual mean concentration and dominant soil types: the denotations are as follows: 1: hilly gully region 1; 2: hilly gully region 2; 3: hilly gully region 3; 4: hilly gully region 4; 5: hilly gully region 5; 6: plateau gullies; 7: terrace; 8: alluvial plain; 9: stony mountains; 10: highland grassland; 11: dry grassland; 12: sandy; 13: hilly woods.

The relationship between annual discharge and concentration is shown in Figure 3R3. As we can see from Figure 3R3a, instead of a bell shape correlation, the mean concentration generally declines with annual discharge for all the 68 study gauges. However, this trend doesn't sustain for the seven gauges at the outlet of major tributaries (Figure 3R3b), where the plot is more scatter. This is consistent with our findings in Figure 1b, that these gauges near the outlet of tributaries have less coupled discharge-concentration relationship. Although the discharge of 1982 – 1986 is smaller than that in 2008 – 2012, we believe the strength of the correlation between Q-C would sustain. Moreover, from Figure 1b, we can see that usually catchments with smaller discharge have stronger Q-C correlation. As we can from Figure 3R3b, the plots are scatter in both 1982 – 1986 and 2008 – 2012 despite of the change in discharge. Thus, we think that the vegetation is a better indicator of concentration than discharge.

Figure 3R3: Scatter plot of annual mean concentration and annual discharge: (a) at 44 study gauges between 1982 and 1986, where the dots are color-coded by the slope in the Q-C regression (ïĄąQC) at each gauge; and (b) at 7 gauges with both data from the years 1982 – 1986 (blue dots) and the years 2008 – 2012 (red dots).

We have added the following explanation in different parts of the manuscript (please see lines 229 – 235 and 348 – 352), we hope the reviewer finds our explanation satisfactory.

L229: To confirm this impact of vegetation resistance, we also examined the relationship between sediment concentration and other catchment characteristic like dominant soil type. No significant correlation was observed as vegetation did. Although there could still be other factors not considered here contributed to the decline in sediment concentration, it is undoubted that vegetation is one of the most influential factors of sediment reduction and can be used as a good indicator of the soil erosion and sediment transport in the YRB.

L348: Numerical simulations as well as long-term measurements on the soil properties are also needed to further explain the physical mechanism of vegetation retardation: how it develops its impact on soil erosion and sediment transport by changing soil properties and other topographic characteristics during its growth and spread.

(4) On line 92-96 about the NDVI data used, how area-specific NDVI was obtained from the spatial imagery? Since the downloaded imagery was global, why only NDVI at 44 gauges not the maximum 68 gauges was estimated.

We extracted the raster data of NDVI from the global image by the drainage area of each gauge. Instead of the 68 gauges, we choose to use the 44 gauges located on the major tributaries for further study, as the water and sediment relationship at gauges on the main stem are more likely to be significantly influenced by the major dames along the YR. The situations on hillslope in the catchments could be overwhelmed by these dam activities. To avoid the significant impact from the human management on water release, we chose these 44 gauges on the major tributaries for our further analysis on the catchment characteristics. We hope the reviewer satisfies with our explanation (please see lines 99 – 101). The following is the explanation we added in the manuscript:

The gauges on the main stem of YR were not used as the water and sediment condition there is more likely controlled by the major dams along the main stem rather than the hillslope characteristics.

(5) Why not use the mean NDVI, but the maximum (daily?) NDVI, when you investigated the relationship between the NDVI and mean concentration at annual scale? How much uncertainty for the maximum NDVI exists?

We tried the mean NDVI as well, the rising and falling trend is still apparent (see following figure), but the maximum NDVI provides better shape. Thus, we chose to use the maximum NDVI for presentation. One possible reason is that the vegetation types in the YRB are mostly deciduous, the green period is relatively short, an averaged NDVI could decrease the difference among vegetation density. Since the variability in maximum NDVI for each site is not very large (Figure 3a), and the trend is consistent between mean NDVI and maximum NDVI, we think the uncertainty for the maximum NDVI is not significant for this study. We hope the reviewer finds our explanation sufficient.

(6) In Figure A1, the text in legend is inappropriate, because there were not 68 green triangles plotted (maybe 68-44=22 gauges).

Since the 44 gauges belongs to the 68 gauges, it might be confusing to use 22, we have changed the symbol to make it clear. Hopefully the reviewer finds the updated figure appropriate.

Figure S1: Spatial distribution of hydrology gauges used in this study. The green triangles correspond to 68 gauges with discharge and sediment concentration data, the red circles correspond to 44 selected gauges with NDVI data, and the yellow circles are the ones with annual discharge and sediment data for the years 2000 – 2012.

(7) On line 120, "annual scale . . . is when the coupling from wavelet coherence analysis is strongest", why?

As we can see from Figure 1a and A3, the annual scale has largest wavelet coherence. The wavelet coherence is like the correlation coefficient, the larger it is, the more correlated the two variables are. Thus, we chose to use the annual for further analysis. We have made the following changes in the manuscript, we hope the reviewer finds our explanation sufficient (please see lines 133 – 135).

This analysis is applied only at annual scale since this is when the coupling from wavelet coherence analysis is the strongest (the one with the largest wavelet coherence).

(8) On line 141-142, "slope in the Q-C regression (aQC) declines exponentially with Qm (p-value < 0.0001)". From the Figure 1b, it looks like log-transformation used to aQC. If so, then the text expression and plot were inconsistent. And the authors should give the mathematical equation of exponential decline trend and its fitted curve.

Thank you for noticing the logarithmic scale in y axis. We chose to use logarithmic scale for the slope in Q-C regression to better present the differences among small values which is a large amount in the study catchment, otherwise the small values would just cluster together. Taking the log-transformation on both side of the exponential regression, we will have a linear relationship between log(ïĄąQC) and Q. We have added the fitted curve and the equation in the updated Figure 1. We are sorry about the confusion and hope the reviewer will find our explanation satisfactory. Thank you!

Figure 1: Scatter plots between long-term mean annual discharge (Qm) and (a) wavelet Q-C coherence at daily, monthly and annual scales, (b) slope of the discharge-sediment concentration regression (ïĄąQC) at annual scale, $R^2$ = 0.55 and p-value < 0.0001.

Please also note the supplement to this comment:
https://www.hydrol-earth-syst-sci-discuss.net/hess-2018-265/hess-2018-265-AC2-supplement.pdf
**Fig. 1.** Figure 2R1: Scatter plots between the maximum NDVI and slope in the Q-C regression (ïĄ̨QC).

[Figure]

**Fig. 2.** Figure 3R1: Scatter plot of annual mean concentration and maximum NDVI: (a) at 44 study gauges between 1982 and 1986, where the dots are color-coded by the slope in the Q-C regression (ïĄąQC) at each ga

[Figure]

**Fig. 3.** Figure 3R2: Scatter plot of annual mean concentration and maximum NDVI: (a) at 44 study gauges between 1982 and 1986, where the dots are color-coded by the slope in the Q-C regression (ïĄąQC) at each ga

[Figure]

**Fig. 4.** Figure R1: Scatter plot of annual mean concentration and dominant soil types: the denotations are as follows: 1: hilly gully region 1; 2: hilly gully region 2; 3: hilly gully region 3; 4: hilly gully

[Figure]

**Fig. 5.** Figure 3R3: Scatter plot of annual mean concentration and annual discharge: (a) at 44 study gauges between 1982 and 1986, where the dots are color-coded by the slope in the Q-C regression (ïĄąQC) at eac

[Figure]

**Fig. 6.** Figure 3R4: Scatter plot of annual mean concentration and mean NDVI: (a) at 44 study gauges between 1982 and 1986, where the dots are color-coded by the slope in the Q-C regression (ïĄąQC) at each gauge

**Legend**

- Hydrology Gauges with 2000s data
- Hydrology Gauges 44
- Hydrology Gauges 68
— Main Stem
— Tributaries
☐ Study Area
☐ Yellow River Basin

**Fig. 7.** Figure S1: Spatial distribution of hydrology gauges used in this study. The green triangles correspond to 68 gauges with discharge and sediment concentration data, the red circles correspond to 44 sel
[Figure]

**Fig. 8.** Figure 1: Scatter plots between long-term mean annual discharge (Qm) and (a) wavelet Q-C coherence at daily, monthly and annual scales, (b) slope of the discharge- sediment concentration regression (ïĄą

---

## Author Response (AR2)

Dear Dr. Tian,

We would like to thank you and the two reviewers for your reviews of our manuscript "Emergent stationarity in Yellow River sediment transport and the underlying shift of dominance: from streamflow to vegetation". We appreciate these insightful inputs that have helped to improve the quality of this manuscript. In response to the comments, we have made corresponding revisions. Our response to each comment is listed below in blue with the specific line numbers of the changes we have made. Again, we appreciate the time and inputs from you and the reviewers.

Best regards,
Sheng Ye,
Qihua Ran,
Xudong Fu,
Chunhong Hu,
Guangqian Wang,
Gary Parker,
Xiuxiu Chen,
Siwei Zhang

Anonymous Referee #1

Accepted as is.

Thank you!

Anonymous Referee #2

Substantial improvement has been made in the revision. However, there are still several minor problems to be answered, which are listed below:

We appreciate the reviewer's comments which help improve our manuscript significantly. We have also made the suggested changes this time. Our response to each comment is listed below in blue with the changes in manuscript, we also include the specific line numbers of the changes we have made.

(1) The notations such as Qa and Qm are suggested to be added in the labels of figures to avoid confusion between them.

We have added the Qa and Qm notation to the labels of Figure 1 and Figure S4. Thank you for pointing this out.

(2) Since most of the results were analyzed among different gauges, that is to say the correlations detected are spatial rather than temporal correlations, the authors may introduce this information in the caption of figures and the corresponding text.

We agree with the reviewer that it helps avoid the potential confusion with the introduction of this information. We have added this in the caption of Figure 1 and 2 as well as the corresponding text (please see lines 141 -142, 145, 151, and 189). We hope the reviewer finds this sufficient.

(3) In Line 94-95 of the revised manuscript, what's the difference between NDVI and LTDR? Why use "instead of "?

NDVI is a vegetation index derived from remote sensing data; while LTDR and GIMMS are data projects storing different products including NDVI. The reason we used the NDVI from LTDR instead of the NDVI from GIMMS is that NDVI from LTDR provides better estimation of the vegetation in the YRB. We are sorry about this confusion due to our writing. We have now revised it (please see line 94), hopefully it is clear now. Thank you!

[revised manuscript text omitted]